# Biological Treatment of Nitroaromatics in Wastewater

Swati Gupta and Zeev Ronen *

Department of Environmental Hydrology and Microbiology, The Zuckerberg Institute for Water Research, The Jacob Blaustein Institutes for Desert Research, Ben-Gurion University of the Negev, Midreshet Ben-Gurion 8499000, Israel; guptas@post.bgu.ac.il
* Correspondence: zeevrone@bgu.ac.il; Tel.: +972-8-659-6895

**Abstract:** Nitroaromatic compounds (NACs), which are widely used in pesticides, explosives, dyes, and pharmaceuticals, include nitrobenzene, nitrotoluenes, nitrophenols, and nitrobenzoates. They are also significant industrial pollutants in the environment. These substances, as well as their derivatives, frequently have toxic or mutagenic properties. Wastewater containing nitroaromatic compounds can be effectively managed by using biological treatment methods that are accessible, cost-effective, and environmentally friendly. This review highlights the latest developments in biological treatment systems for removing NACs from wastewater. The large-scale implementation of biological treatment systems will be facilitated by future studies that focus on identifying the best operational methods and that determine how co-pollutants impact the removal of NACs from wastewater.

**Keywords:** nitroaromatic compounds; bioremediation; nitrophenols; aerobic degradation; nitrotoluenes; nitrobenzoates

## 1. Introduction

Nitroaromatic compounds (NACs) are a family of organic contaminants comprising an aromatic ring with an attached nitro group. NACs, which include nitrobenzene, nitrotoluenes, nitrophenols, and nitrobenzoates, are highly significant in the industrial domain and can be used as raw material to create various chemicals, including dyes, pharmaceuticals, explosives, cosmetics, pesticides, and herbicides [1–3]. Nitro groups are essential components of many biologically active molecules, including antibiotics and medications [4]. The primary source of NACs is human-produced environmental pollutants. However, some microorganisms, plants, and fungi produce them as secondary metabolites. NACs, with an estimated $10^8$ tons produced annually, are important industrially because of the diverse chemistry of the nitro group [5]. During manufacture and use, NACs are released into the environment, resulting in severe environmental pollution, thus threatening humans and other living organisms as they contaminate rivers and wastewater. For instance, nitrobenzene can lead to many health issues, including anemia, methemoglobinemia, skin irritation, and liver damage [2,6,7]. Nevertheless, the annual release of nitrobenzene into natural waterways may exceed 9000 metric tons [8]. Additionally, chlorinated nitroaromatic compounds (CNAs) are human-made substances utilized in producing industrial chemicals, dyes, and pharmaceuticals. Because of their hematotoxicity, genotoxicity, and immunotoxicity, CNAs are hazardous to humans and animals [9,10]. p-chloronitrobenzene (p-CNB), commonly used in various industries, is extremely hazardous and may have carcinogenic and mutagenic effects due to its electrophilic nitro and chlorine groups [11]. According to reports, industrial wastewater contains p-CNB at concentrations as high as 200 mg/L [12]. The chemical structures of some nitroaromatic compounds are shown in Figure 1.

Dinitrotoluenes are crucial raw materials in the chemical industry. 2,4-dinitrotoluene (2,4-DNT) is a recalcitrant compound with strongly toxic biological impacts. 2,4-DNT contamination is a global issue caused by widespread explosives production and military

operations [13,14]. Another NAC, 2,4,6-trinitrotoluene (TNT), is the most persistent environmental pollutant. Annual TNT production is estimated to be close to $10^6$ kg. In several countries, including the US, Israel, Germany, and Canada, TNT soil and groundwater pollution has become a severe issue [15,16]. During the transformation process, TNT and its metabolites can produce toxic waste. Moreover, other NACs, such as mononitrobenzoates, are frequently used in manufacturing plastics, elastomers, and polyurethane foams. Furthermore, because of their mutagenic and genotoxic characteristics, they have been found to have harmful impacts on different organisms [4,17]. An important NAC chemical, 4-nitrophenol (4-NP), is utilized in the large-scale production of acetaminophen, an aspirin alternative, and pesticide manufacture [18]. 4-NP has been classified as a hazardous, persistent pollutant. Several studies have been conducted on its carcinogenicity and toxicity [19]. Nitrophenols are hazardous to plants and many other organisms and may accumulate in the food chain [20]. In Punjab, India, 4-NT was found in agricultural run-off and industrial wastewater at 749 and 913 ng/mL [21].

Another study found that different types of nitrobenzoic acids, nitrobenzyl alcohols, and nitrophenols were present in leachate and groundwater in Germany [22]. Furthermore, the products or the altered intermediates produced during the reduction of nitro groups are associated with NAC toxicity. Some intermediates, such as the hydroxyl amino derivatives, are highly toxic and can easily interact with proteins or DNA to cause mutagenesis. The leading causes of nitroaromatic chemical toxicity include electron transfer, reactive organic species, and oxidative stress [23–25]. In brief, when NACs are reduced by one electron, free radicals are produced that are then scavenged by oxygen, producing superoxide radicals and oxidative stress (Figure 2) [7]. The inhalation, skin contact, and direct consumption of NACs can result in methemoglobinemia, nausea, anemia, respiratory distress, and headache. The US Environmental Protection Agency (USEPA) has listed several NACs as "primary pollutants". Most are hazardous, mutagenic, and can cause cancer with prolonged exposure [2]. Consequently, NACs have drawn global attention due to the difficulty in effectively removing these compounds from the environment. They are classified as recalcitrant pollutants due to their electron shortage, which can be increased by the presence of multiple nitro groups or their combination with other electron-withdrawing groups [7,26]. Removing NACs from contaminated sites is challenging because they are stable and hazardous. Therefore, a thorough strategy is needed to detoxify these harmful compounds.

NACs in wastewater have been removed using various chemical (advanced oxidation processes and hydrolysis) and physical (ultrafiltration, incineration, volatilization, photooxidation, and adsorption) techniques. However, these methods are inefficient and not cost effective, and the contaminants that remain after treatment persists as pollutants or toxic substances that require additional attention [5,7,27,29–31]. Bioremediation, among other alternative methods, is gaining popularity worldwide for cleaning up NAC-contaminated sites because it utilizes microorganisms to degrade these recalcitrant compounds. For many years, industrial and municipal pollutants have been treated with microorganisms. Biological treatment is believed to be less expensive and easier to implement [2,5,32]. Bacteria, algae, yeasts, and fungi can degrade NAC contaminants in the environment. Numerous studies have employed an organic co-substrate to accelerate the metabolism of nitroaromatic explosives because they are resistant to oxidation [15]. Surprisingly, microorganisms have been identified that can utilize a variety of NACs as nitrogen and carbon sources for growth, such as nitrotoluene, nitrobenzene, chloronitrobenzenes, and nitrobenzoates [1]. In this review, we provide appropriate suggestions that can assist in directing future studies on this topic.

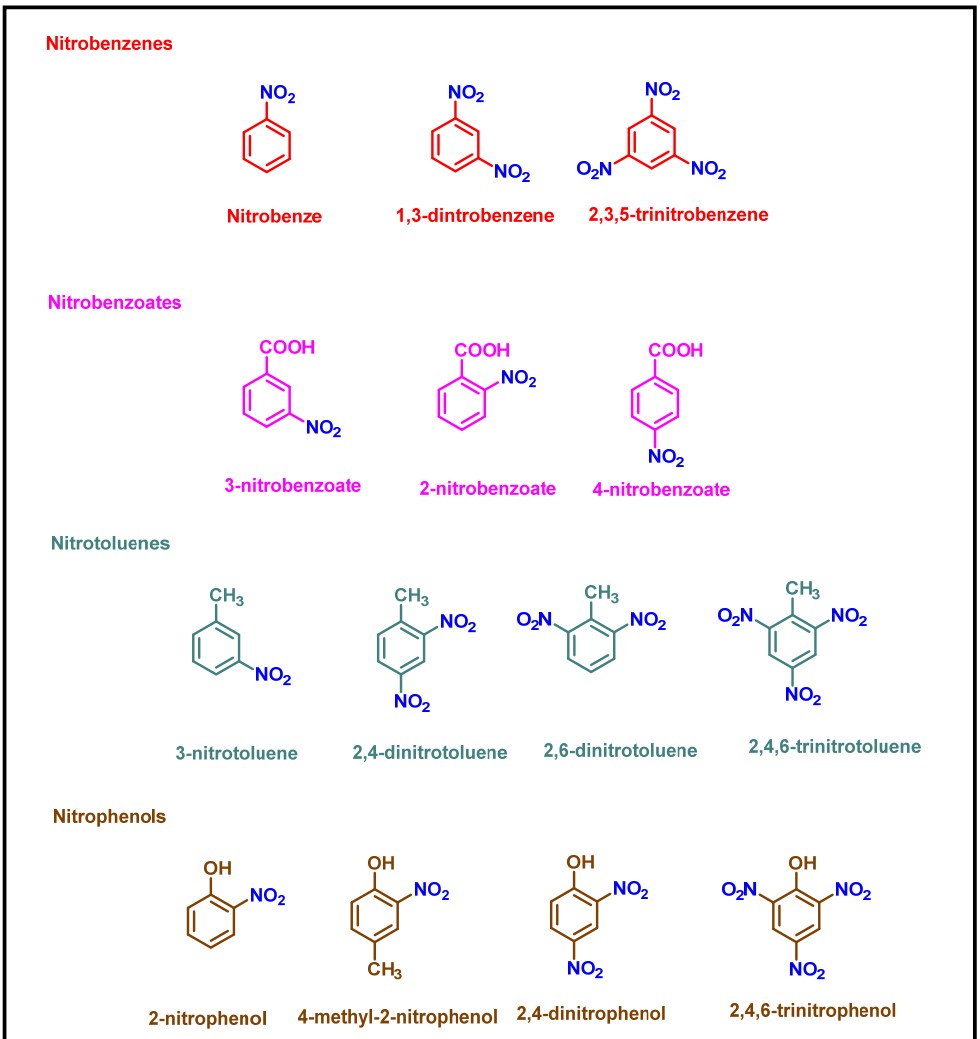

**Figure 1.** Structures of some common nitroaromatic compounds [7,9,26–28].

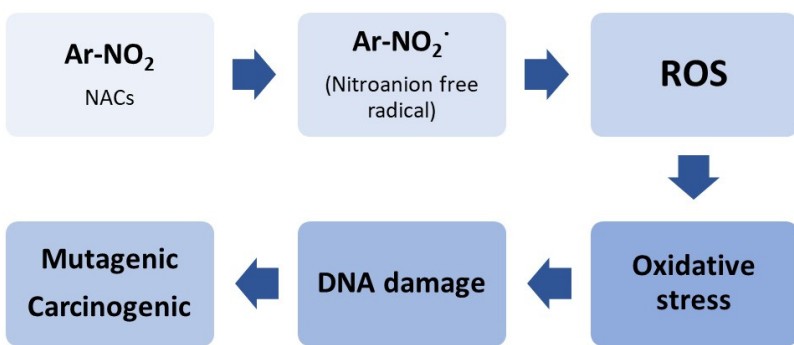

**Figure 2.** Mechanism of NAC toxicity [7,24].

## 2. Bioremediation of NACs

### 2.1. Bacterial Degradation of NACs under Aerobic Conditions

The use of bacteria to clean up polluted environments is growing and developing significantly. Nitroaromatic compounds are degraded by specific bacteria through their metabolic processes [20,33], which can be carried out either anaerobically or aerobically, frequently using certain enzymes, including nitroreductases, monooxygenases, and dioxygenases, for NAC degradation [5,27,34]. The number of nitro groups on the aromatic ring can affect the extent of these compounds' bacterial degradation. The aromatic ring becomes

more electron-deficient as the number of nitro groups increases, contributing to resistance to oxidative degradation by dioxygenases. However, poly-nitroaromatic compounds, such as 2,4,6-trinitrophenol and 2,4,6-trinitrotoluenes, remain effective in reductive degradation by nitro reductases in aerobic conditions [15,35–37].

Nevertheless, mono- and dinitrotoluene molecules undergo oxygenase reactions. In order to deal with the nitro group in an aerobic environment, bacteria have developed four main approaches: deoxygenation (dioxygenases insert two -OH groups on the NAC aromatic ring and release the nitro group as nitrite simultaneously), monooxygenation, partial reduction of the nitro group to hydroxylamine (which hydrolyzes to produce ammonia) using NADH as a reducing equivalents; and the formation of a hydride-Meisenheimer complex (following nitrite release, the complex rearomatizes) [5,38]. This section discusses the bacterial degradation of nitrobenzene, nitrotoluenes, nitrobenzoate, and nitrophenols.

A number of bacteria that can utilize nitrobenzene (NB) as a source of carbon and energy have been identified and reported by researchers. Two main pathways are involved in aerobic NB degradation: the partial reduction of the nitro group and the widely studied dioxygenase mechanism [26]. Under aerobic conditions, *Comamonas* sp. strain JS765 can use nitrobenzene as the sole source of nitrogen, carbon, and energy. A multicomponent nitrobenzene dioxygenase (NBDO) system has been discovered in this strain. Nitrobenzene dioxygenase (NBDO) is a member of the naphthalene family of multicomponent Rieske-type dioxygenases [1]. When NBDO adds both dioxygen atoms to a nitroaromatic ring, a dihydroxy intermediate is produced that spontaneously rearranges to produce a catechol and release nitrite [39–41]. Dioxygenases catalyze ring-opening processes when aromatic hydrocarbons undergo oxygenation reactions to create molecules with at least two hydroxyl groups. To be mineralized, these metabolites eventually enter the tricarboxylic acid cycle (Figure 3) [27].

NB is also degraded by *Serratia* sp. NB2, *Arthrobacter* sp. NB1, and *Stenotrophomonas* sp. NB3. These three strains have been isolated from polluted sludge. When used as the only nitrogen and carbon source, mixed cultivation shows better degradation than mono-cultivation at an initial concentration of 400 mg/L [42]. A unique strain of halophilic bacteria, called *Bacillus licheniformis* strain YX2, has been shown to degrade nitrobenzene. The NB degradation pathway by YX2 is a partial reductive pathway [29], indicated by the accumulation of the metabolite 2-aminophenol and ammonia release. It was discovered that *Pseudomonas frederiksbergensis* strain NB-1 can successfully degrade nitrobenzene. NB-1 showed the greatest adaptive range in temperature (4–35 °C) and pH (5–11) [43]. It has been reported that certain bacterial strains can also degrade 1,3-dinitrobenzene and 1,3,5-trinitrobenzene via dioxygenation and nitro group reduction (Figure 3) [23].

Nitrophenols (NPs) are widely used in pharmaceuticals, dyes, and insecticides [26]. Large amounts of nitrophenols have been released into the environment because of inappropriate waste disposal methods, agricultural usage, and medical applications. These substances have been found in rainwater, surface water, groundwater, and industrial effluents [28]. It has been reported that numerous bacteria can degrade nitrophenols, and some of these bacteria can use nitrophenols as their only source of carbon and energy. Two primary mechanisms for the microbial degradation of 4NP have been proposed: (i) the hydroquinone pathway (HQ pathway), and (ii) the nitrocatechol pathway (NC pathway). Several Gram-negative bacteria possess well-characterized HQ pathways. There have been reports of several Gram-positive and some Gram-negative bacteria exhibiting the NC route [44]. *Arthrobacter* sp. strain CN2, isolated from activated sludge, can degrade 4-nitrophenol. During 4-nitrophenol's degradation by strain CN2, the authors identified three metabolites (1,2,4-benzenetriol, 4-nitrocatechol, and maleylacetate). Following the release of nitrite from 4-nitrocatechol, 1,2,4-benzenetriol is produced. According to earlier reports, these metabolites are a part of the 1,2,4-benzenetriol (BT) or 4-NC pathway [45]. 4-NP is also degraded by *P. putida* 1274, releasing nitrite. Hydroquinone was the primary degraded intermediate, which is part of the HQ pathway (Figure 4) [46].

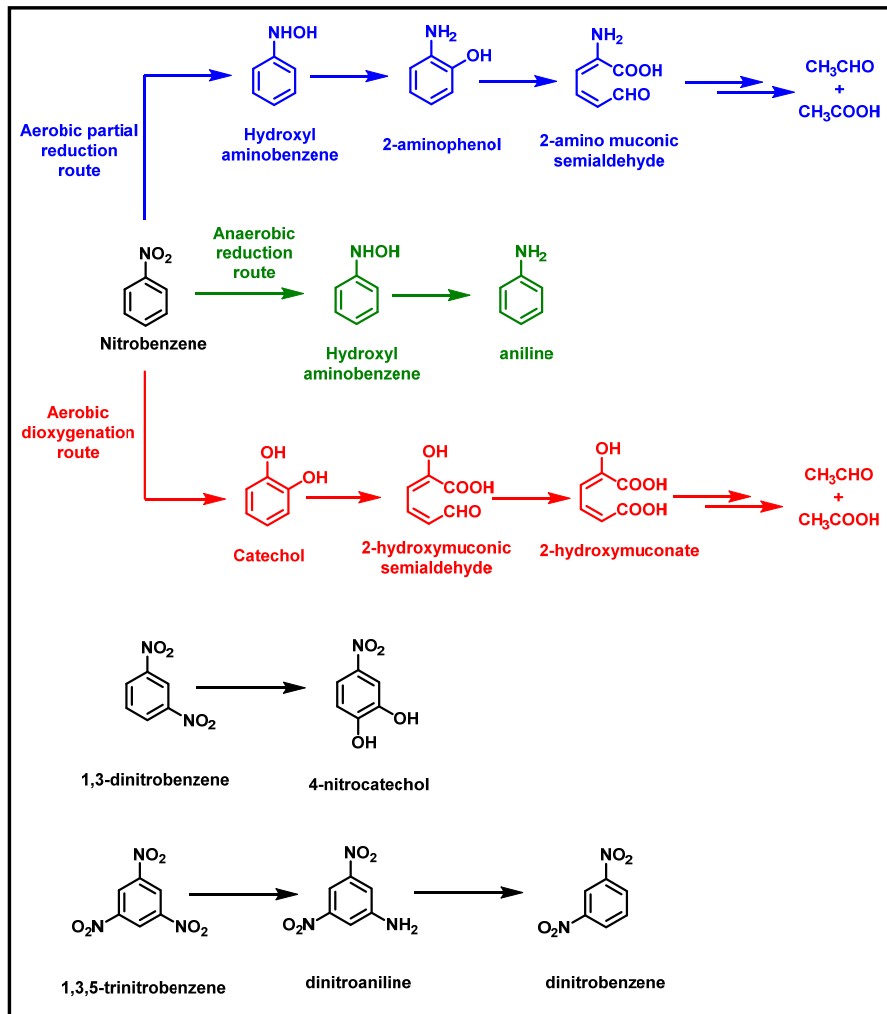

**Figure 3.** Bacterial degradation pathway of nitrobenzenes under aerobic conditions; adapted from [1,23,26,27].

The most straightforward procedure for mononitrophenol degradation is observed in 2-nitrophenol (2-NP). Nitrite and catechol are produced when a monooxygenase attacks the 2-NP in *Pseudomonas putida* B2 [26]. It was discovered that *Alcaligenessp* strain NyZ215 can also degrade 2-nitrophenol using a catechol ortho-cleavage route. This strain can utilize 2-nitrophenol as the only source of nitrogen, carbon, and energy [47]. *Cupriavidus necator* JMP134 is able to grow on 3-nitrophenol and use it as its only energy, nitrogen, and carbon source [48]. There are two known pathways for the metabolism of 3-nitrophenol: one through 1,2,4-benzenetriol (formed after releasing ammonia from hydroxylaminophenol) in *Pseudomonas putida* B2, and the other through aminohydroquinone in *Cupriavidus necator* JMP134. Both pathways begin with a reductase and reduce 3-nitrophenol to hydroxy-laminophenol (Figure 4) [1,48].

**Figure 4.** Bacterial degradation pathway of mononitrophenols under aerobic conditions [28,45–48].

Several strains capable of growing on 2,6-dinitrophenol, 2,4-dinitrophenol, or 2,4,6-trinitrophenol (picric acid) have been isolated. For example, 2,6-DNP can be broken down by *Cupriavidus necator* JMP134 and *Pseudomonas* sp. N-26-8, converting it into 2-hydroxy-5-nitromuconate (HNM) via 4-nitropyrogallol (produced after releasing nitrite from 2,6-DNT). 2-hydroxy-5 nitropenta-2,4-dienoic acid was produced by the spontaneous decarboxylation of the cleavage product HNM (Figure 5) [1,28]. *Burkholderia* KU-46 can degrade 2,4DNP by forming 1,4-benzoquinone, 4-nitrophenol (NP), and nitrite [49]. *Anabaena variabilis*, a cyanobacterium, showed a strong capacity for 2,4-DNP degradation. It removes 2,4-DNP by transforming it to 2-amino-4-nitrophenol (2-ANP) [50]. *Rhodococcus* sp. B1 degraded 2,4-DNP, releasing a nitro group from its ortho-position and producing 3-nitroadipic acid [28]. Because of their nitro groups, NACs have an electron deficiency, making electrophilic attacks difficult and preventing aromatic biodegradation. Therefore, polynitroaromatics, such as 2,4,6-trinitrophenol (TNP), undergo preliminary reductive transformations [51,52]. Using an NADPH-dependent reductase with cofactor F420, picric acid can be reduced to a hydride complex, forming 2,4-dinitrophenol and releasing nitrite. Ultimately, 2,4-dinitrophenol is hydrolytically cleaved to 4,6-dinitrohexanoate via a second reduction and a hydride-Meisenheimer complex (Figure 5) [53,54].

**Figure 5.** Bacterial degradation pathway of di- and trinitrophenols under aerobic conditions [1,28,49,55].

Numerous bacterial strains that may use nitrotoluenes as a carbon and nitrogen source have been isolated from different sources. *Diaphorobacter* strain DS2 is one of the nitrotoluene degraders that can use 3-nitrotoluene as its carbon and nitrogen source. 3-nitrotoluene dioxygenase, in this strain, transforms 3-NT into 3-methylcatechol, which is subsequently degraded by the meta-cleavage mechanism. The degradation of nitrobenzene and other isomers of mononitrotoluene (4-nitrotoluene (4-NT) and 2-nitrotoluene (2-NT)) have also been observed in strain DS2 [40,56]. Another bacterial strain, *Rhodococcus* strain ZWL3NT, degrades 3-NT via 3-methyl catechol, which is then further oxidized by ortho- and meta-cleavage routes [57]. Using a meta-cleavage route, *Micrococcus* strain SMN-1 degraded 2-nitrotoluene to 3-methylcatechol and released nitrite (Figure 6) [58]. Different bacteria, such as *Pseudomonas* strain 4NT, *P. putida* TW3, *Mycobacterium* strain HL-4NT-1, and *Acidovorax* strains JS42-KSJ9, 10, and 11, degrade 4-nitrotoluene via different pathways [59].

**Figure 6.** Bacterial degradation pathway of mononitrotoluenes under aerobic conditions [56,58,59].

Various bacterial strains have been shown to degrade 2,4-dinitrotoluene (2,4-DNT). Using monooxygenase and dioxygenase enzymes, *Burkholderia* sp. strains DNT and R34 degrade 2,4-DNT. Initially, the benzene ring is attacked by dinitrotoluene dioxygenase (DNTDO), which converts DNT into 4-methyl-5-nitrocatachol (4M5NC) and simultaneously removes a nitrite group. Moreover, MNC monooxygenase transforms the substrate into 2-hydroxy-5-methylquinone by eliminating another nitrite, which results in 2,4,5-trihydroxytoluene (Figure 7) [13,38,60]. *Hydrogenophaga palleronii* strain JS863 and *Burkholderia cepacia* strain JS850 can utilize 2,6-DNT as a source of nitrogen and carbon. First, 2,6-DNT undergoes dioxygenation to transform into 3-methyl-4-nitrocatachol (3M4NC), which is followed by nitrite release. Subsequently, without releasing the second nitro group before the ring cleavage, 3M4NC undergoes meta-ring cleavage (Figure 7) [1,61].

It is difficult for bacteria to degrade TNT [62]. There are three nitro groups in the aromatic ring of TNT. Consequently, it is susceptible to reductive degradation but resistant to oxidative degradation by aerobic bacteria [63]. In many recent publications, TNT degradation has been shown to be caused by the reductive mechanism [64]. *Diaphorobacter* strain DS2 can utilize TNT as a source of nitrogen and carbon. In strain DS2, nitroreductase partially reduces TNT to 2-amino-4,6-dinitrotoluene and 4-amino-2,6-dinitronitrotoluene (2-ADNT and 4-ADNT), which are then dioxygenated by multi-component dioxygenase systems to complete TNT degradation in aerobic conditions [15]. *Stenotrophomonas* strain SG1, and *Rhodococcus* strain YH1 can transform TNT via a reductive pathway. The basic mechanism by which aerobic bacteria transform TNT to amino dinitrotoluene often results in reduced dead-end products, which can then react with one another to generate azotetranitrotoluene. Another TNT degradation pathway is a hydride addition to the benzene ring, which produces mono- and dihydride-Meisenheimer complexes (Figure 7) [36,37,65].

**Figure 7.** Bacterial degradation pathway of di and trinitrotoluenes under aerobic conditions [15,36–38].

Nitrobenzoates are degraded by bacteria using either the oxidative or reductive pathways. In the reductive process, the first step is a nitroreductase-catalyzed process that releases ammonium, while in the oxidative process, the nitro group is removed as nitrite by oxygenase. The reductive pathway is the only way that bacterial 2-nitrobenzoate (2NB) and 4-nitrobenzoate (4NB) degradations have so far been observed [1,26]. *Arthrobacter* sp. SPG metabolized 2NB by accumulating nitrite ions. During the degradation of 2NB, two metabolites (salicylate and catechol) were identified (Figure 8). The 2NB degradation process investigated in strain SPG differs from all other reported bacterial 2NB degradation pathways [17]. Prior research indicates that 2NB bacterial breakdown happens reductively through ammonium ion accumulation; however, strain SPG breaks down 2NB oxidatively by releasing nitrite ions [17,66]. The genus *Cupriavidus* strain ST-14 was identified as having the ability to use 2NB and 4NB for its growth. 2NB transforms into 3-hydroxyanthranilate, and 4-nitrobenzoate transforms into protocatechuate by this strain (Figure 8). By adding 3-nitrobenzoate dioxygenase, the recombinant strain ST-14 can use 3-nitrobenzoate (3NB) through the protocatechuate metabolic pathway, which makes it easier for one bacterial strain to break down all three mononitrobenzoate isomers [67–69].

**Figure 8.** Bacterial degradation pathway of nitrobenzoates under aerobic conditions [67–69].

## 2.2. Bacterial Degradation of NACs under Anaerobic Conditions

The reactions of nitroaromatic compounds in anaerobic systems almost exclusively involve reducing the nitro group. Under anaerobic conditions, nitroreductase catalyzes the complete reduction of nitro groups to amino groups via hydroxyamino intermediates. There are two types of nitroreductases: type I (oxygen insensitive) and type II (oxygen-sensitive) [5,26]. Many studies have demonstrated that the reduction of NACs' nitro group to an amino group is the first step in their biodegradation (Figure 9) [2,14,27,28,70]. Mc-Cormick et al. [71] demonstrated that *Vieilonella alkalescens* cell extract could reduce 40 mono-, di-, and trinitroaromatic compounds with hydrogen as an electron donor, and the reactivity of the nitro groups depended on other substituents and the position of the nitro groups relative to these substituents. Liu et al. [72] examined the ability of *Shewanella oneidensis* MR-1 (electrochemically active bacteria) to reduce 2,6-DNT anaerobically. During anaerobic respiration on 2,6-DNT, strain MR-1 obtained energy for growth. *Stenotrophomonas* strain SG1 can transform TNT in the presence of citrate and nitrate under anaerobic conditions. During the transformation process, SG1 reduced TNT to aminodinitrotoluene isomers [36]. When 2,4-DNT is biotransformed by *Shewanella marisflavi* EP1 in anaerobic environments, lactate can be used as an electron donor by EP1 to dissimilatory reduce 2,4-DNT. Two intermediates, 2-amino-4-nitrotoluene and 4-amino-2-nitrotoluene, are initially formed during the 2,4-DNT transformation, leading to the final product, 2,4-diaminotoluene [14].

Furthermore, under sulfate-reducing conditions, TNT was metabolized by *Desulfovibrio* sp. (B strain) within ten days, with pyruvate as the primary substrate, sulfate as the electron acceptor, and TNT as the sole nitrogen source. Under nitrogen-limiting conditions, 100% removal of TNT was observed. The central intermediate followed was diaminonitrotoluene, which was converted to toluene via triaminotoluene by a reductive deamination process [73].

**Figure 9.** Bacterial degradation pathways of some nitroaromatic compounds under anaerobic conditions [36,74].

In conclusion, anaerobic reduction of nitro groups to nitroso derivatives, hydroxylamines, or amines occurs through the successive addition of electron pairs donated by co-substrates. The advantage of the anaerobic process is that most poly- nitroaromatics are susceptible to biodegradation only under these conditions. Individual species of anaerobic microbes rarely bring about the complete conversion of the nitroaromatic substrate to $CO_2$ or methane. Hence, synergistic participation of a consortium of bacteria is required for partial/complete degradation of the compounds. Aromatic amines have a higher biodegradability and less toxicity. Consequently, a practical approach to breaking down NACs is to pretreat the contaminated wastewater anaerobically before the aerobic biological treatment.

### 2.3. Degradation of NACs by Fungi, Yeast, and Algae

The lignin-degrading fungus *Phanerochaete chrysosporium* was investigated for its ability to degrade a range of nitroaromatic compounds. Several metabolic intermediates were identified from 4-nitrotoluene (4-NT). 4-nitrobenzyl alcohol (4-NBA) was initially produced from 4-NT, and then 4-nitrobenzaldehyde and 4-nitrobenzoic acid were formed through oxidation reactions. *Phanerochaete chrysosporium* also transformed 2- and 3-nitrotoluenes to the corresponding aryl alcohols [75]. Under ligninolytic conditions, 4-nitrophenol (4-NP)

was degraded by *Phanerochaete chrysosporium* into 1,2-dimethoxy-4-nitrobenzene via the intermediate 4-nitroanisole. The methylation of the phenolic hydroxyl group and hydroxylation of aromatic rings were involved in this metabolic pathway [76]. The white-rot fungus Trametes versicolor strain BAFC 2234 can also degrade 4-nitrophenol [77]. The fungal strains that can degrade TNT through their metabolic pathways include Phanerochaete, Micromycetes, Stropharia, Gymnophilus, Kuehneromyces, Pseudoarthrobacter, and Basidiomycetes [2,78]. The ligninolytic enzyme produced by white-rot fungi, consisting primarily of peroxidases and laccases, catalyzes TNT degradation [79].

Yeast strains may be suitable for NAC bioremediation. *Yarrowia lipolytica* can transform TNT through the principal process of aromatic ring reduction, which is mediated by hydride ions, and a minor nitro group reduction [80]. Another yeast strain, *Rhodotorula mucilaginosa* Z1, can degrade nitrobenzene in aerobic conditions [81]. Microalgae provide a promising bioremediation option by converting carbon, nitrogen, phosphorus, and synthetic organic compounds into biomass. Khromykh et al. investigated the capacity of the genus *Rhizoclonium* (common freshwater algae) to detoxify TNT in designed water pollution conditions. The obtained results demonstrate the effectiveness of the biodegradation process and the potential of *Rhizoclonium* algae as a cleaning agent for TNT-contaminated water bodies [82]. Static experiments examined the bio-reaction between *Microcystis aeruginosa* and nitrobenzene (NB) at varying NB concentrations and initial algal densities. The bio-reaction was supported by high algal densities and high initial NB concentrations [83].

Algae produce or break down nitroaromatic compounds based on different environmental conditions. *Chlorella pyrenoidosa* reacts differently to NACs depending on the type of stress. While the rate of 4-nitroaniline (4-NA) and 4-nitrophenol (4-NP) degradation is accelerated in the presence of NACs, particularly in starvation conditions, increased exposure to 4-NA leads to an increase in 4-NA secretion from the algae [84]. Under photoautotrophic conditions, some microalgae have been shown to degrade NACs. Effective NP degradation in wastewater was reported to be achieved by *Anabaena variabilis*, *Chlorella vulgaris*, and *Chlorella fusca* [85]. TNT in seawater can be naturally removed by three species of marine macroalgae: two red (*Porphyra yezoensis* and *P. hornemannii*) and one green (*Acrosiphonia coalita*). All three macroalgae convert TNT to 2-ADNT and 4-ADNT [86].

## 3. Biological Treatment Methods for NAC Removal from Wastewater

### 3.1. Activated Sludge Process

The activated sludge process (ASP) is one of the most well-established and widely used methods for biologically treating wastewater from industrial sewage. The process involves treating wastewater with microorganisms in an aerobic reactor. The treated wastewater is removed from the microorganism-containing sludge as the microorganisms break down the pollutants, and the sludge is frequently recycled back into the reactor [87]. Nitrogen-containing organic pollutants were effectively removed by the bioaugmented activated sludge generated by inoculating *Comamonas* sp. Z1 and *Acinetobacte* sp. JW. According to this study, bioaugmentation may help remove organic pollutants that contain nitrogen, resulting in a potentially useful wastewater treatment method [88]. In a slurry bubble column, it was discovered that the waste-activated sludge was highly efficient at aerobically breaking down 4-nitrophenol (4-NP) at higher 4-NP concentrations. Furthermore, the amount of dissolved oxygen (DO), which is regulated by the rates of aeration and oxygen uptake, has a significant impact on the 4-NP aerobic biodegradation rate [89,90]. The disadvantage of the ASP system is that it requires a large setup area and often produces low-quality effluent that fails to fulfill micropollutant discharge standards.

### 3.2. Sequencing Batch Reactor

The sequencing batch reactor (SBR), an updated ASP system, uses the same reactor to separate biomass from treated wastewater. Nitrobenzene degradation by aerobic granular sludge has been reported so far. This sludge, cultivated in SBR reactors, degraded nitrobenzene as a carbon and nitrogen source. Nitrobenzene concentrations as high as 600 mg/L did

not inhibit this process [91]. By sustaining a high enough biomass to quickly convert DNP to non-toxic stages and extending aeration to avoid the formation of hazardous byproducts, activated sludge adapts to toxic concentrations of DNP [92]. In wastewater with a high ammonia nitrogen content, 4-NP may be eliminated by nitrifying bacteria in an SBR [93]. In NAC treatment, an anaerobic–aerobic SBR that uses sludge from an anaerobic wastewater treatment plant as an inoculum functions similarly to one that uses sludge from a primarily aerobic municipal wastewater treatment plant [94]. An aerobic sludge SBR is an effective method of treating wastewater containing NACs. Comparing a bioaugmented SBR to a non-bioaugmented SBR, the bioaugmentation containing 4-NP-degrading bacteria improved the removal of 4-NP, obtaining continuous 4-NP shock loads [95]. 4-NP inhibited aerobic granule formation and caused pre-cultivated granules to disintegrate when it was the sole carbon source. However, the functional and structural strength of pre-cultivated aerobic granular sludge was maintained when it was co-substrated with acetate, indicating the significance of a beneficial carbon source for the efficient degradation of toxic wastes [96]. Zhu et al. [97] showed that using anaerobic bioreactors, 2,4-dinitrophenol and nitrate could be successfully removed from synthetic wastewater. The inocula from anaerobic bioreactors intended to remove nitrate and phenol facilitated the simultaneous breakdown of these pollutants [97]. The benefits of SBR systems include their compact size, high treatment effectiveness, and potential for anaerobic–aerobic activities in a single reactor. The primary drawbacks include a highly variable functional timetable and a complex operational strategy [87].

### 3.3. Biofilm-Based Reactor

Microorganisms, such as fungi, bacteria, and protozoa, adhere to filter material to form a biofilm, which is then used to absorb organic pollutants from influent wastewater and purify the water [27]. Microbial cells are wrapped in a matrix of extracellular polymeric substances (EPSs) to form biofilms, essential for pollutant migration and transformation. The constituents of EPSs include proteins, polysaccharides, and other polymers. The functional groups in these substances enable them to bind with various heavy metals and organic contaminants [98]. Biofilm systems, such as moving-bed biofilm reactors (MBBRs) or fixed-bed biofilm reactors (FBBRs), have shown promise in recent years as a way to remove pollutants from wastewater [99]. Lu et al. [98] demonstrated EPS's important function in mediating the transfer of NB and protecting biofilms from NB-induced damage by binding interactions, mainly involving amino and carboxyl groups. EPSs also play a major role in influencing the removal of NB and denitrifying biofilms' responses to NB stress [98]. Ji et al. [100] investigated the biodegradation of continuously loaded 4-NP in an anaerobic biological fluidized bed (ABFB) reactor, achieving 99% 4-NP removal. The microbial community characteristics in aerobic sludge changed significantly, as evidenced by the notable differences in EPS contents and microbial components [100].

High 4-NP concentrations (100–400 mg/L) were treated using a lab-scale membrane-aerated biofilm reactor (MABR) system. The MABR achieved high removal efficiencies of 4-NP and COD, demonstrating its efficiency and reliability [101]. The biodegradation of 3-nitrophenol (3-NP) was investigated by Gonzalez et al. [102] in a continuous upflow fixed-bed biofilm reactor (FBBR). Microorganisms that degrade 3-NP were utilized to develop an efficient continuous biofilm reactor for the degradation of m-nitrophenol [102]. The impact of biofilm carriers on biofilm formation and 4-nitrophenol (4-NP) degradation was investigated using vertical baffled biofilm reactors (VBBR) outfitted with Fe and plastic carriers. The findings demonstrated that Fe carriers accelerated 4-NP degradation and biofilm formation. 4-NP entered the citrate cycle mostly via acetyl-CoA after transforming via 4-nitrocatechol and 1,2,4-benzenetriol [103]. In the VBBR, simultaneous removal of nitrobenzene (100 μM), aniline (100 μM), and nitrate (1000 μM) was accomplished in less than eight hours. The coexistence of almost incompatible reduction and oxidation steps in a single VBBR was demonstrated by the reduction of nitrate and nitrobenzene and the mineralization of aniline [104].

In wastewater treatment, a biological aerated filter (BAF) acts as an immobilization reactor, providing high-quality effluent and resistance to organic or toxic loads. BAFs are excellent at eliminating nitrogen and organic matter from wastewater. In BAFs, rough porous spherical filter materials are commonly used to give microorganisms a favorable growth environment because they can improve adherence to biofilms [105,106]. In a continually operating BAF (filled with porous ceramic particles and attached immobilized cells), the study demonstrated the microbial purification of synthetic wastewater with TNP, yielding low residual COD and TNP concentrations. TNP overloading, on the other hand, inhibited nitrite-oxidizing activity, which reduced TNP degradation efficiency in the BAF system [107]. An immobilized microbial process, comprising an anaerobic and biological aerated filter (I-AF-BAF) system, was used to break down TNT in an aqueous solution. The process succeeded in effectively degrading TNT. The main anaerobic degradation products that were found included 2-amino-4,6-dinitrotoluene, 4-amino-2,6-dinitrotoluene, 2,4-diamino-6-nitrotoluene, and 2,4-diamino-6-nitrotoluene [108]. An innovative membrane-aerated biofilter (MABF) was developed to treat nitroaniline wastewater contaminated with 4-nitroaniline (4-NA) and 2-nitroaniline (2-NA). The conversion and removal of NA were aided by the presence of the co-metabolic substrate acetic acid. Compared to 2-NA, 4-NA had a more significant removal loading and was more biodegradable [105].

*3.4. Upflow Anaerobic Sludge Blanket Reactor*

Anaerobic wastewater treatment methods, such as the upflow anaerobic sludge blanket (UASB) reactor, have advantages over other anaerobic and aerobic techniques. Utilizing microorganisms that form granular structures inside the reactor, the UASB reactor effectively treats wastewater and produces biogas [109]. Full-scale UASB reactors have been used to treat a range of wastewater because they tolerate moderate variations in pH, temperature, and wastewater composition [87]. Xu et al. investigated the reduction of 4-nitrophenol (4-NP) by co-metabolism in an anaerobic UASB reactor. The main product of 4-NP reduction was 4-aminophenol (4-AP), which did not further degrade in the reactor. Three co-substrates (glucose, methanol, and sodium acetate) were tested to enhance the 4-NP reduction in the reactor; of these, sodium acetate performed best [110]. In two lab-scale UASB reactors, She et al. [20] examined the removal efficiencies of 2,6-dinitrophenol (2,6-DNP), using two distinct co-substrates. The anaerobic sludge utilized sodium acetate and glucose as co-substrates to detoxify and degrade 2,6-DNP. On the other hand, systems that were fed glucose had greater inhibition from 2,6-DNP at high amounts than systems fed sodium acetate [111]. The study examined the effects of the $COD/NO_3$-N ratio on the biotransformation and removal of 2-NP, 4-NP, and 2,4-DNP in bench-scale UASB reactors [112]. The main intermediate metabolites were 2-AP, 4-AP, and 2-A,4-NP. In addition, the reactors showed simultaneous methanogenesis and denitrification [112].

In a lab-scale UASB reactor, She et al. [113] investigated the degradation of 3-NP using sodium acetate as a co-substrate. The highest concentration of 3-NP found in the effluent was 71.6 mg/L. The primary intermediate product of the degradation process of 3-NP was 3-AP [113]. The disadvantages of UASB reactors include their very long initiation period and their difficult recovery from high-stress levels [87]. The UASB and a staged multi-phase anaerobic reactor can be combined in the anaerobic baffled reactor (ABR), a novel third-generation anaerobic reactor setup. Its effectiveness was evaluated while NB was acclimated in an ABR. The chemical oxygen demand and NB removal efficiencies were 90% and 98%, respectively [114]. After comparing the anaerobic sequencing batch reactor (ASBR) and the UASB processes, the anaerobic migrating blanket reactor (AMBR) was developed [115]. Kuscu and Sponza's study [116] demonstrated that 4-NP-containing synthetic wastewater can be successfully treated in an AMBR reactor at various hydraulic retention times (HRTs), ranging from 1 to 10.38 days. Between 92% and 94% of the COD and 4-NP removal efficiencies were maintained for HRTs of 2.4 and 10.38 days [116].

### 3.5. Membrane Bioreactor

The membrane bioreactor (MBR), which combines a bioreactor and a separation membrane, has been viewed as an appealing solution to address the drawbacks of conventional activated sludge processes, including biological instability, low sludge quality, and low concentration of suspended mixed liquor solids [117]. The study in [118] assessed nitrophenol biodegradation in a continuous stirred-tank reactor (CSTR-MBR) and modified Ludzack–Ettinger MBR (MLE-MBR) systems. The removal efficiencies for nitrophenols varied from 87% to 96%, while the removal efficiencies for total inorganic nitrogen were 52% and 75% for the CSTR-MBR and MLE-MBR, respectively. Both MBRs showed similar preferences for NP biodegradation. In the CSTR-MBR, in contrast with MLE-MBR, metabolic uncoupling by nitrophenols resulted in a lower ATP yield and less bound EPS, indicating better membrane fouling control in the CSTR-MBR system [118]. To optimize 4-NP biodegradation, two integrated membrane-aerated bioreactor systems (RA and RB) were designed with anoxic and aerated zones. 4-NP in RA was first degraded anaerobically in the anoxic zone; next, 4-AP was formed and then degraded aerobically in the aerated zone. 4-NP was first broken down aerobically in RB, which resulted in nitro group release and further breakdown pathways involving the β-ketoadipate in the TCA cycle [119]. NB removal was significantly aided by activated sludge. Significant improvements in the NB removal efficiency were obtained by extending the solid retention time in the MBR from 2 to 25 days. A beneficial correlation was also seen between the biomass content and the NB removal efficiency [120]. A list of bioreactor systems is shown in Table 1.

### 3.6. Microbial Fuel Cells

Microbial fuel cell (MFC) systems comprise microorganisms that use a series of reactions to transfer electrons from an organic or inorganic compound's oxidation product to ATP, creating an electrical current [121]. MFCs can break down or stabilize contaminants found in wastewater. Over this past decade, MFCs that produce electricity or eliminate contaminants have drawn much attention [122]. Electrochemically active microorganisms transfer electrons from a donor to an electrode in an MFC. These electrons are then transferred to the cathode via an external circuit, which reduces the final electron acceptor [123]. The anode and cathode operate in anaerobic and aerobic conditions.

Given that the MFC anode can act as a sufficient anaerobic terminal electron acceptor for microbes, the degradation of organic pollutants is improved over traditional anaerobic reactors [124]. 4-nitrophenol (4-NP) has been degraded in an MFC using electrochemical technology. The anode biofilm could break down different aromatic compounds. Furthermore, the study of the microbial community composition revealed that the anodic biofilm was primarily populated by functional bacteria [125].

High NB tolerance and NB removal, alongside simultaneous electricity production, were demonstrated by single-chamber microbial fuel cells (S-MFCs) with activated carbon (AC) air cathodes and bio-anodes. The S-MFCs showed excellent results in removing NB, with an efficiency of over 97% [126]. When oxygen was added to an anode MFC, *Pseudomonas monteilii* LZU-3 broke down 4-nitrophenol (4-NP) and produced current. The study showed that adding more oxygen to the anode MFC could be useful for producing electricity and biodegrading recalcitrant substances such as NACs [127]. Mu et al. [123] investigated nitrobenzene elimination using a bioelectrochemical system (BES) with microbial acetate oxidation at the anode and nitrobenzene reduction at the cathode. The main byproduct of nitrobenzene reduction in the BES was aniline, and BES's current density affected both the removal and formation of aniline at different rates [123]. A sulfate-reducer-enriched biocathode in the MEC was utilized to achieve effective NB reduction. Aniline and sulfide were the primary reduced products, with respective conversion efficiencies of 97% and 78%. Since no sulfate was present in the catholyte, the high NB removal efficiency indicated that it might be utilized as an electron acceptor for generating electricity by the cathodic biofilm [128].

### 3.7. Constructed Wetlands

Constructed wetlands (CWs) constitute biological techniques that have gained global attention for their ability to clean up different wastewaters. CWs are designed systems, built and established to make use of constructed and naturally occurring processes to filter out various pollutants from wastewater. There are two primary types of wetlands: subsurface flow CWs and surface flow CWs. Based on the surface or subsurface flow direction, systems are classified as horizontal or vertical [129]. The efficiency of horizontal subsurface flow-constructed wetlands (HSSFCWs) in nitrobenzene degradation was examined by Kirui et al. [130]. The study was conducted using two lab-scale wetlands planted with Juncus effusus. One wetland had constant aeration, while the other remained unaerated. Based on the findings, both wetlands performed at 99% NB removal overall. It has been suggested that detailed microbial reactions involving NB as an electron donor and sulfate as an electron acceptor produce high concentrations of generated sulfide [130]. Regardless of external carbon sources, a study on three CW groups (glucose, starch, and blank CW group) showed effective removal of NB and COD in wastewater with a low NB concentration (<50 mg/L). However, treating wastewater with high NB concentrations in CWs required using suitable external carbon sources, which accelerated the conversion of NB to AN and decreased toxicity to microorganisms and plants [131].

To simultaneously treat wastewater containing nitrobenzene and produce electricity, a single-chamber membrane-less air-cathode microbial-fuel-cell-linked constructed wetland (MFC-CW) was erected [132]. The MFC-CW was able to generate power more efficiently than an MFC. Nitrobenzene degradation is aided by a suitable hydraulic retention time (HRT) and concentration ratio of NB to COD. Additionally, co-metabolism caused by microorganisms and glucose may accelerate NB reduction [132]. Di et al. [133] examined the impact of radial oxygen loss (ROL) from various plants (*Scirpus validus, Typha orientalis, Iris pseudacorus*) on the generation of bioelectricity in constructed wetland–microbial fuel cells (CW-MFCs) and the treatment of wastewater containing NB. Scirpus validus demonstrated resistance to NB, which resulted in higher COD and NB removal efficiency and a significant amount of bioelectric production in the CW-MFC [133].

In conclusion, wastewater treatment (WWT) can use several different options to remove NACs. The ability of sequential batch reactors (SBRs) to modify operations in response to the wastewater's particular characteristics makes them especially effective at addressing NACs, in addition to other contaminants. The advantages of SBR systems include significant treatment effectiveness and the potential for anaerobic–aerobic operations in the same reactor [27]. The primary drawbacks are the complicated operational plan and the significant fluctuations in operational duration. MBR treatment of NAC-polluted water shows good efficacy; however, increasing expenses and membrane fouling problems prevent MBR use from becoming widespread, especially in high-organic-pollutant environments. The growth of attached biomass in the biofilm process makes it more resistant to NAC loading compared to SBR. The most developed and effective method of WWT in removing NACs is the use of anaerobic reactors. Anaerobic reactors, such as anaerobic membrane bioreactors (AMBRs) and anaerobic baffled reactors (ABRs), provide a number of benefits, including the ability to produce biogas for future use and to manage high NAC concentrations, and they possess strong removal capabilities. MFCs have the ability to degrade NACs with remarkable efficiency due to their high microbial activity, and they can also generate electricity in the process.

Our thorough literature review has revealed a wealth of evidence supporting the effectiveness of biological treatment in removing NACs. However, it is concerning that we could not find any evidence of full-scale implementation for wastewater treatment. It is imperative that we take action and scale up the use of this proven method to ensure the safe and effective treatment of wastewater.

**Table 1.** Bioreactors for NAC removal.

| Bioreactor | Biomass | NACs | Condition | Carbon Source | NAC Removal Efficiency (%) | Degradation Products | Ref. |
|---|---|---|---|---|---|---|---|
| Slurry bubble column | Activated sludge | 4-NP | Aerobic | None | 100% | - | [90] |
| SBR | Granular sludge | NB | Aerobic | None | 100% | $CO_2$ | [91] |
| SBR | Activated sludge | 4-NP | Aerobic | Glucose | 100% | $CO_2$ | [95,134] |
| SBR | Granular sludge | 4-NP | Aerobic | Acetate | 100% | $CO_2$ | [96] |
| SBR | *Rhodococcus opacus* strain JW01 | TNP | Aerobic | Glucose | >99.9% | $CO_2$ | [135] |
| SBR | Nitrifying sludge | 4-NP | Aerobic | Glucose | 99.9% | - | [93] |
| UASB | Anaerobic sludge | 4-NP | Anaerobic | Sodium acetate | 96–100 | 4-AP | [110] |
| UASB | Granular sludge | 2,6-DNP | Anaerobic | Glucose, sodium acetate | >98% | - | [111] |
| UASB | Anaerobic granular sludge | 2-NP, 4-NP, 2,4-DNP | Anaerobic | Sodium acetate | >99% | 2-AP, 4-AP, 2-A,4-NP | [112] |
| UASB | Anaerobic sludge | 3-NP | Anaerobic | Sodium acetate | >95% | 3-AP | [113] |
| ABR | Anaerobic granular sludge | 4-NP | Anaerobic | Glucose | 99% | 4-AP, phenol, ammonia | [136] |
| AMBR | Anaerobic granular sludge | 4-NP | Anaerobic | Glucose | 94% | 4-AP | [116] |
| Packed bed biofilm reactor | *Arthrobacter chlorophenolicus* A6 | 4-NP | Aerobic | None | 100% | $CO_2$ | [137] |
| Biological aerated filter (BAF) reactor | *Rhodococcus* sp. NJUST16 | TNP | Aerobic | None | 98% | $CO_2$ | [107] |
| BAF and anaerobic filter (AF) | B925 | TNT | Anaerobic– aerobic | Ethanol | 100% | $CO_2$ | [108] |
| Aerobic biological fluidized bed (ABFB) reactor | Activated sludge | 4-NP | Aerobic | None | 99% | $CO_2$ | [100] |
| Membrane-aerated biofilm reactor (MBR) | Activated sludge | 4-NP | Aerobic | None | 94.40% | $CO_2$ | [101] |
| Anaerobic semi-fixed bed biofilm reactor | Anaerobic digestion sludge | 4-NP | Anaerobic | Glucose | 98% | 4-AP | [138] |
| MBR | Activated sludge | 4-NP | Anoxic– aerobic | None | 95.86% | $CO_2$ | [119] |

## 4. Conclusions

In conclusion, biological treatment techniques provide suitable and environmentally friendly methods to handle wastewater containing nitroaromatic compounds. Several biological treatment technologies have been discussed in this review, such as membrane bioreactors, microbial fuel cells, activated sludge processes, sequencing batch reactors,

biofilm processes, anaerobic reactors, and constructed wetlands. Every method has its benefits, but problems still need to be solved. These include establishing the most effective way to operate in large-scale applications. Additionally, it is necessary to understand the effects of co-pollutants on NAC removal from wastewater. The continued investigation of microbial-based treatments holds great potential for reducing the environmental impact of nitroaromatic compounds, even though more developments and research are required to overcome current constraints. To sum up, an in-depth understanding of the microbial metabolism of NACs is essential for developing, refining, and implementing efficient wastewater treatment strategies. Using this information, treatment procedures can be designed to minimize expenses, energy usage, and environmental impacts while achieving high levels of NAC degradation. For sustainable solutions to the problems of NAC pollution in wastewater to be advanced, more research and innovation in this area are necessary.

Nevertheless, industries have yet to widely adopt the biological treatment of nitroaromatics in wastewater for several reasons. The difficulties include the absence of effective full-scale mineralization technologies, uncertainty surrounding performance optimization, and our limited understanding of biotransformation mechanisms. To close these gaps, biotechnologists could be essential in conducting additional research to clarify biotransformation mechanisms, enhance treatment efficacy, and create adaptable technologies. To complete these knowledge gaps, remove challenges, and enable the successful application of biological treatment techniques for nitroaromatic compounds, industry and academia need to collaborate.

**Author Contributions:** Conceptualization, Z.R.; writing—original draft preparation, S.G.; writing—review and editing, Z.R.; visualization, S.G.; funding acquisition, Z.R. All authors have read and agreed to the published version of the manuscript.

**Funding:** This research was funded in part by the Israel Science Foundation (ISF) grant number 2712/17.

**Data Availability Statement:** No new data were created or analyzed in this study. Data sharing is not applicable to this article.

**Acknowledgments:** We would like to thank Samara Bel for reviewing the manuscript.

**Conflicts of Interest:** The authors declare no conflicts of interest.

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
