# Peer review of "Biological Treatment of Nitroaromatics in Wastewater"

_water, doi:10.3390/w16060901_

Round 1
Reviewer 1 Report
Comments and Suggestions for Authors
This manuscript attempts to review biological treatment of nitroromatics in wastewater.
Unfortunately there is not enough attention paid to anaerobic pathways and methodology. Most of the text and all of the Figures are on aerobic processes and pathways. The advantages and disadvantages of the two approaches should be discussed specifically and systematically. It seems that the current figures are all for aerobic pathways, this should be made explicit in the figure titles. The aerobic degradation routes seems to contain reductive steps. The authors should make clear how this is possible in aerobic systems/cells and indicate how the reducing equivalents needed for these steps are generated. In addition the authors should be clear about the “N products” generated in the various steps (nitrate?, nitrite?, NO2?, NH3?, you name it.).
The authors should also explicitly review to what extent biological treatment/removal of nitroromatics is carried out and achieved at full scale, and the types of reactors / systems currently used.
Also, the authors should pay attention to their writing style.
As an example: line 228-229: “aromatic ring structure consisting of three nitro groups” ? Really? “oxygenase degradation by aerobic bacteria” of course? “makes it susceptible to reductive degradation” why is that? In the subsequent lines the authors present a reductive pathway in an aerobe; will be counterintuitive to many readers and should be discussed explicitly.
2.2: only 10 lines on anaerobic degradation routes?!
Conclusions: to what extent is biological treatment of nitroromatics in wastewater applied by industry / what precludes acceptance of the technology? Is there a role for biotechnologist to show the way? Which way looks best? Or is that too early to say? Discuss and make knowledge gaps/roadblocks and how to overcome them explicit!
Comments on the Quality of English Languageas indicated above: be careful how you phrase things!
Author Response
Dear Editor,
We want to express our gratitude to the reviewers for providing us with their valuable feedback. We have carefully considered their comments and have made the necessary revisions to improve the paper. Please find enclosed a detailed response to the reviewer, along with an explanation of the revisions we have made. We believe that the suggested revisions have significantly enhanced the quality of the paper, and we kindly request you to consider publishing it in Water.
Reviewer 2
This manuscript attempts to review biological treatment of nitroromatics in wastewater.
1.Unfortunately there is not enough attention paid to anaerobic pathways and methodology. Most of the text and all of the Figures are on aerobic processes and pathways. The advantages and disadvantages of the two approaches should be discussed specifically and systematically. It seems that the current figures are all for aerobic pathways, this should be made explicit in the figure titles. The aerobic degradation routes seems to contain reductive steps. The authors should make clear how this is possible in aerobic systems/cells and indicate how the reducing equivalents needed for these steps are generated. In addition the authors should be clear about the “N products” generated in the various steps (nitrate?, nitrite?, NO2?, NH3?, you name it.).
Reply: We thank the reviewer comment. We extended our description of anaerobic metabolism of NAC and section 2.2 revised accordingly. The figures captions were corrected. We also explain that under anaerobic conditions, the nitro group of Nitroaromatic compounds is reduced to amino group (line 284-294). Nevertheless, aerobic bacteria can also reduce the nitro group of nitroaromatic compounds by utilizing NADH as an electron donor (line 122-123). We also noted that aromatic amines are less toxic and more biodegradable. Thus, pretreating anaerobically prior to beginning an aerobic treatment is an efficient way to break down NACs. For N products, we have mentioned it in text (line 135, 146, 168, 170, 174, 180, 192, 195, 197-198, 203, 218).
2.The authors should also explicitly review to what extent biological treatment/removal of nitroromatics is carried out and achieved at full scale, and the types of reactors / systems currently used.
Reply: Although it is well known that nitro-aromatic compounds have the potential to degrade in contaminated areas, we could not find in the scientific literature reports on the full-scale operational mineralization technology for nitro-aromatic compounds. The majority of investigations have been conducted on a lab scale. The list of the reactors has been shown in table 1.
3.Also, the authors should pay attention to their writing style. As an example: line 228-229: “aromatic ring structure consisting of three nitro groups” ? Really? “oxygenase degradation by aerobic bacteria” of course? “makes it susceptible to reductive degradation” why is that? In the subsequent lines the authors present a reductive pathway in an aerobe; will be counterintuitive to many readers and should be discussed explicitly.
Reply : Thanks for the comment We have modified the sentence and stated that “There are three nitro groups in the aromatic ring of TNT. Consequently, it is susceptible to reductive degradation but resistant to oxidative degradation by aerobic bacteria” (line 239-241).
4.only 10 lines on anaerobic degradation routes?!
Reply: Thanks for the comment. We have extended our discussion on anerobic degradation (line 277-316).
Conclusions: to what extent is biological treatment of nitroromatics in wastewater applied by industry / what precludes acceptance of the technology? Is there a role for biotechnologist to show the way? Which way looks best? Or is that too early to say? Discuss and make knowledge gaps/roadblocks and how to overcome them explicit!
Reply We added the following text in the conclusions: Industries have yet to widely adopt the biological treatment of nitroaromatics in wastewater for several reasons. The difficulties include the absence of effective full-scale mineralization technologies, uncertainty surrounding performance optimization, and our limited understanding of biotransformation mechanisms. To close these gaps, biotechnologists could be essential in conducting additional research to clarify biotransformation mechanisms, enhance treatment efficacy, and create adaptable technologies. To complete these knowledge gaps, remove challenges, and enable the successful application of biological treatment techniques for nitroaromatic compounds, industry and academia need to collaborate. (line 619-627)
Reviewer 2 Report
Comments and Suggestions for Authors
Dear authors,
please address the following points:
1. Introduction. Line 64. Please use the same concentration units (mg/L, µg/L, etc) throughout the text. The same applies to concentrations on line 391.
2. Line 159. "HQ pathway and the (ii) NC pathway" Please, explain briefly these mechanisms at this point.
3. I guess that PNP and 4-NP refer to para-nitrophenol. If this is correct, please use the same acronym. 4-NP has been defined again on lines 422 and 493 please correct this.
4. Line 380. Please indicate how high were those concentrations (in mg/L).
5. Lines 436-438. The sentence is not clear. I guess that those results appear at very high concentrations of 3NP in the influent so that microbial inhibition was produced. If this is Ok indicate which were those concentrations. Please, clarify.
6. The advantages expressed in Lines 419-420 and on: "Full-scale UASB reactors have been used to treat a range of wastewater because they tolerate variations in pH, temperature, and wastewater composition [85]." seem to contradict those of 438-440: "The disadvantages of UASB reactors include ... and their difficult recovery from high stress levels". Please, re-write to avoid confusion. I suggest just adding "moderate variations in pH..."
7. Lines 481-482. In the sentence "The anode and cathode operate in anaerobic and aerobic conditions." Please indicate if the electrodes can interchange conditions or always work in the same conditions.
8. Lines 516-519. It would interesting to indicate if both CWs (aerated and non aerated) achieved the same removals considering the energy expense and aerobic conditions in each case. Additionally, di sulfate reduction to sulfide occurred in both CWs?, because in aerobic conditions that reaction is inhibited.
9. Line 538. NACs has been already defined.
Author Response
Dear Editor,
We want to express our gratitude to the reviewers for providing us with their valuable feedback. We have carefully considered their comments and have made the necessary revisions to improve the paper. Please find enclosed a detailed response to the reviewer, along with an explanation of the revisions we have made. We believe that the suggested revisions have significantly enhanced the quality of the paper, and we kindly request you to consider publishing it in Water.
Reviewer 3
- Introduction. Line 64. Please use the same concentration units (mg/L, µg/L, etc) throughout the text. The same applies to concentrations on line 391.
Reply: Thanks for the comment, we have changed the units accordingly (line 65, 411).
- Line 159. "HQ pathway and the (ii) NC pathway" Please, explain briefly these mechanisms at this point.
Reply: Thanks for the comment, we have modified it (line 163-165). The bacterial degradation of 4NP has been occurred via hydroquinone pathway (HQ pathway) and the (ii) via nitro catechol pathway (NC pathway).
- I guess that PNP and 4-NP refer to para-nitrophenol. If this is correct, please use the same acronym. 4-NP has been defined again on lines 422 and 493 please correct this.
Reply: Thanks for the comment, we have corrected the text. PNP and 4-NP refer to para-nitrophenol. We use 4-NP for para-nitrophenol in the text (line 512).
- Line 380. Please indicate how high were those concentrations (in mg/L).
Reply: Thanks for the comment, we have added the information in the text (line 421).
- Lines 436-438. The sentence is not clear. I guess that those results appear at very high concentrations of 3NP in the influent so that microbial inhibition was produced. If this is Ok indicate which were those concentrations. Please, clarify.
Reply: Thanks for the comment, we have modified the text (line 477-80).
- The advantages expressed in Lines 419-420 and on: "Full-scale UASB reactors have been used to treat a range of wastewater because they tolerate variations in pH, temperature, and wastewater composition [85]." seem to contradict those of 438-440: "The disadvantages of UASB reactors include ... and their difficult recovery from high stress levels". Please, re-write to avoid confusion. I suggest just adding "moderate variations in pH..."
Reply : We have modified the text (line 461-463).
- Lines 481-482. In the sentence "The anode and cathode operate in anaerobic and aerobic conditions." Please indicate if the electrodes can interchange conditions or always work in the same conditions.
Reply : To our knowledge, they can be interchanged. Additionally, MFC can operate independently in a variety of anodic environments, such as completely anaerobic, aerobic, and aerating environments. (reference: https://doi.org/10.1016/j.biortech.2019.03.130)
- Lines 516-519. It would interesting to indicate if both CWs (aerated and non aerated) achieved the same removals considering the energy expense and aerobic conditions in each case. Additionally, di sulfate reduction to sulfide occurred in both CWs?, because in aerobic conditions that reaction is inhibited.
Reply : Both CWs show 99% NB removal. Sulfide concentration was about 84 mg/L in an unaerated wetland. However, the concentration of sulfide in the continuously aerated wetland was less than 0.5 mg/L.
- Line 538. NACs has been already defined.
Reply : We have changed it (line 557).
Round 2
Reviewer 1 Report
Comments and Suggestions for Authors
Reviewer 2 point 2: The authors should explicitly state this in the manuscript.
Comments on the Quality of English LanguageReviewer 2 point 2: The authors should explicitly state this in the manuscript.
Author Response
Dear Editor,
We would like to express our gratitude to the reviewers for providing us with their valuable feedback. We have carefully considered their comments and have made the necessary revisions to improve the paper. Please find enclosed a detailed response to the reviewer, along with an explanation of the revisions we have made. We believe that the suggested revisions have significantly enhanced the quality of the paper, and we kindly request you to consider publishing it in Water.
2. The authors should also explicitly review to what extent biological treatment/removal of nitroromatics is carried out and achieved at full scale, and the types of reactors/systems currently used.
Reply: Although it is well known that nitro-aromatic compounds have the potential to degrade in contaminated areas, we could not find in the scientific literature reports on the full-scale operational mineralization technology for nitro-aromatic compounds. The majority of investigations have been conducted on a lab scale. The list of reactors is shown in Table 1. We explicitly stated (596-600) that "Our thorough literature review has revealed a wealth of evidence supporting the effectiveness of biological treatment in removing NACs. However, it is concerning that we could not find any evidence of full-scale implementation for wastewater treatment. It is imperative that we take action and scale up the use of this proven method to ensure the safe and effective treatment of wastewater."
Reviewer 2 Report
Comments and Suggestions for Authors
The manuscript can be accepted.
Author Response
We thank the reviewer for his valuable comments.